# Surveillance of Daughter Micronodule Formation Is a Key Factor for Vaccine Evaluation Using Experimental Infection Models of Tuberculosis in Macaques

**DOI:** 10.3390/pathogens12020236

**Published:** 2023-02-02

**Authors:** Isabel Nogueira, Martí Català, Andrew D. White, Sally A Sharpe, Jordi Bechini, Clara Prats, Cristina Vilaplana, Pere-Joan Cardona

**Affiliations:** 1Radiology Department, ‘Germans Trias i Pujol’ University Hospital, 08916 Badalona, Spain; 2Comparative Medicine and Bioimage Centre of Catalonia (CMCiB), Germans Trias i Pujol Research Institute (IGTP), 08916 Badalona, Spain; 3Escola d’Enginyeria Agroalimentària i de Biosistemes de Barcelona Departament de Física, Universitat Politècnica de Catalunya (UPC)-BarcelonaTech, 08860 Castelldefels, Spain; 4UK Health Security Agency, Porton Down, Salisbury SP4 0JG, UK; 5Unitat de Tuberculosi Experimental, Germans Trias i Pujol Research Institute (IGTP), 08916 Badalona, Spain; 6Centro de Investigación Biomédica en Red de Enfermedades Respiratorias (CIBERES), 28029 Madrid, Spain; 7Direcció Clínica Territorial de Malalties Infeccioses i Salut Internacional de Gerència Territorial Metropolitana Nord, 08916 Badalona, Spain; 8Microbiology Department, North Metropolitan Clinical Laboratory, ‘Germans Trias i Pujol’ University Hospital, 08916 Badalona, Spain; 9Genetics and Microbiology Department, Universitat Autònoma de Barcelona, 08913 Cerdanyola del Vallès, Spain

**Keywords:** tuberculosis, BCG vaccine, aerosol vaccination, non-human primate, macaque, bubble model, computed tomography scanner

## Abstract

Tuberculosis (TB) is still a major worldwide health problem and models using non-human primates (NHP) provide the most relevant approach for vaccine testing. In this study, we analysed CT images collected from cynomolgus and rhesus macaques following exposure to ultra-low dose *Mycobacterium tuberculosis* (Mtb) aerosols, and monitored them for 16 weeks to evaluate the impact of prior intradermal or inhaled BCG vaccination on the progression of lung disease. All lesions found (2553) were classified according to their size and we subclassified small micronodules (<4.4 mm) as ‘isolated’, or as ‘daughter’, when they were in contact with consolidation (described as lesions ≥ 4.5 mm). Our data link the higher capacity to contain Mtb infection in cynomolgus with the reduced incidence of daughter micronodules, thus avoiding the development of consolidated lesions and their consequent enlargement and evolution to cavitation. In the case of rhesus, intradermal vaccination has a higher capacity to reduce the formation of daughter micronodules. This study supports the ‘Bubble Model’ defined with the C3HBe/FeJ mice and proposes a new method to evaluate outcomes in experimental models of TB in NHP based on CT images, which would fit a future machine learning approach to evaluate new vaccines.

## 1. Introduction

Understanding the dynamics of tuberculosis (TB) disease is essential for the design of better strategies against it, as TB is a cause of high morbidity and mortality worldwide. *Mycobacterium tuberculosis* (Mtb) infects about 10 million people every year and a quarter of humanity is estimated to be infected. Moreover, it has been the deadliest single infectious disease for decades, killing nearly 1.5 million people annually [1]. Unfortunately, this mortality has been overtaken by SARS-CoV-2 in 2020, a further consequence of which has been a devastating impact on TB care resulting in an estimated additional half a million TB deaths, thus going back to a mortality level similar to that of 2010 [2,3]. Defeating TB is not yet a feasible goal in the short term, but endeavours must continue to focus on the three cornerstones of prevention, early diagnosis and treatment, and advance the development of new vaccines, biomarkers and drugs with shortened protocols [4,5,6].

The granuloma is the hallmark of the host response against Mtb infection, appearing as a characteristic round histologic structure and ranging in size from 1–3 mm in diameter [7,8,9,10,11,12]. There are several studies that describe granuloma subtypes, mainly based on their cell proliferation, matrix quality and pattern distribution under a microscope [9,10,11,13,14,15,16]. In the context of image analysis, granulomas are expected to profusely mineralize and thus be identifiable as calcified micronodules on CT scan images collected from macaques mainly from 20 weeks after experimental infection [11].

Vaccination is one of the essentials for controlling the disease. The Bacille Calmette–Guérin (BCG) vaccine is an attenuated strain of *Mycobacterium bovis*, which was first used in 1921 and still plays a pivotal role in high endemic TB countries, despite low efficacy in some regions [17,18]. BCG has demonstrated protection against TB death and disseminated disease, especially against childhood tuberculous meningitis and miliary disease, but has also shown disparate results in the protection afforded to adults and pulmonary involvement [19,20,21,22,23,24]. Even though new vaccine candidates are in development [25,26,27], the centenary vaccine remains as the only licensed vaccine [1] and clinical trials are currently ongoing to explore the efficacy of BCG novel protocols, such as the use of alternative delivery routes such as inhaled (or aerosolized) [28,29,30,31,32], or in revaccination regimens [33,34].

The investigation of prevention strategies and treatments depends on well-characterized experimental animal models. The most clinically relevant experimental model is provided by non-human primates (NHP) as they resemble humans in almost every aspect and mimic human physiological patterns. Macaques are widely reported to manifest the same TB disease as people do [35], also sharing comparable efficacy after BCG vaccination and chemotherapy [36,37,38,39]. It is worth emphasizing that the investigation of BCG vaccination in macaques allows the exploration of the whole spectrum of the protection status seen in humans, as the vaccine confers a protective effect in cynomolgus macaques and limited protection in rhesus [28,29,40,41]. The model in rhesus macaques is the one that robustly reproduces active TB lesions [14,36,42,43,44,45]. In contrast, cynomolgus macaques reveal a greater similarity with the wide spectrum of outcomes that follow human-Mtb interaction, especially the Asian cynomolgus (with Indonesian and Chinese genotypes), which are better able to contain the progression of the infection, reflecting mainly a latent TB infection, and more limited progression towards active disease [14,15,29]. However, Mauritanian genotype cynomolgus macaques are more susceptible to TB disease, showing outcomes of TB infection more akin to those found in rhesus macaques [45,46].

One important consideration when designing experimental models is to use the route of natural infection and an optimal challenge dose that is sufficient to induce measurable disease, but not at a level that could overwhelm potentially protective effects induced by vaccination and confound the evaluation assessment [14,15,30,36,38,44,47]. The infection after experimental exposure to aerosols containing ultra-low dose inoculum has been demonstrated to closely resemble naturally occurring Mtb exposure, causing disease-induced changes in the lung that can be monitored [14,48]. The evaluation method should be sensitive to allow the detection of subtle differences among prophylactic candidates.

TB disease dynamics are not yet entirely understood. In fact, there is still no canonical hypothesis to explain the progression from latent to active disease. Daughter lesions have already been described during active TB development, appearing near larger lesions, and being associated with the advance of local infection. The ‘Bubble Model’, firstly conceived by Prats et al. [49] and inspired by soap bubble formation together with previous concepts [5,49,50,51,52,53], hypothesized that progression to active TB was the consequence of the intensity of the inflammatory response in the initial micronodules, the induction of daughter micronodules around the initial nodule and the coalescence of all these lesions that cause the formation of a consolidated lesion, with cavitation capacity. However, this model was built on data obtained from the murine model with C3HeB/FeJ mice by Marzo et al. [54]. Supportive data obtained from the NHP model would endorse future confirmation in humans.

Image analysis plays a fundamental role in TB disease assessment. Chest X-ray was classically used for diagnosis and still remains a basic tool due to its widespread availability and sufficient sensitivity. Recently, high-quality diagnostic images such as computed tomography (CT), positron emission tomography–computed tomography (PET-CT) and magnetic resonance imaging (MRI) have emerged. However, to our knowledge, there have been limited opportunities to characterize the development of Mtb-induced disease, particularly the early events after infection. Scanning enables the collection of high-quality data in a non-invasive way, allowing the examination of features of particular interest such as lung consolidations, nodules, micronodules (daughter and isolated), cavitation, tree-in-bud lesions and their distribution [5,11,46,48,49,54,55,56,57,58,59,60,61,62]. CT scanning allows the acquisition of high-quality images in a few seconds (notably faster than MRI and PET-CT), thus minimizing interference caused by respiratory motion, and being also the most available and economical three-dimensional imaging technology. A score system for the estimation of TB burden disease in the NHP model based on CT images developed by Sharpe et al. [63] has been reported.

The aim of this study was to evaluate the progression of Mtb infection in experimental NHP models through the spatial monitoring of the micronodules. We found that the identification of consolidations associated with daughter micronodules would be a useful tool to evaluate the efficacy of new vaccines.

## 2. Materials and Methods

### 2.1. Animals

Images were obtained from 35 macaques aged 3–4 years; 13 cynomolgus macaques (*Macaca fascicularis*, of Indonesian genotype) and 22 rhesus macaques (*Macaca mulatta*, of Indian genotype). The macaques enrolled in the studies were obtained from established closed breeding colonies in the United Kingdom and absence of previous exposure to mycobacterial antigens was confirmed by tuberculin skin test as part of colony management procedures and by screening for IFN-γ ELISPOT (MabTech, Nacka, Sweden) responses to tuberculin-PPD (SSI, Copenhagen, Denmark), and pooled 15-mer peptides of ESAT6 and CFP10 (Peptide Protein Research LTD, Fareham, UK). BCG vaccinations were delivered to sedated animals either as a 100 μL intradermal (ID) injection using Danish strain 1331 (SSI, Copenhagen, Denmark) delivered to the upper left arm, or by exposure to aerosolised BCG Danish strain 1331 created using an Omron MicroAir mesh nebuliser (Omron Healthcare UK Ltd., Milton Keynes, UK) and a modified paediatric anaesthesia mask. The BCG vaccine was prepared for intradermal administration according to manufacturer’s instructions for administration to humans. One ml of Sautons diluent was added to a vial of BCG vaccine to give a suspension of BCG at an estimated concentration of between 2 and 8 × 10^6^ CFU/mL. The vaccination dose was selected to be equivalent to a standard adult intradermal dose [13,14,28]. The macaques were experimentally infected with Mtb (Erdman strain K 01) ultra-low dose aerosol exposure (presented dose 19–26 CFU) as previously described [14], and following exposure to MTB, macaques were housed in facilities compliant with the Advisory Committee for Dangerous Pathogens Level 3 (ACDP3) regulations. Compatible social groups were housed in accordance with the Home Office (UK) Code of Practice for the Housing and Care of Animals Used in Scientific Procedures (1989), and the National Committee for Refinement, Reduction and Replacement (NC3Rs) Guidelines on Primate Accommodation, Care and Use, August 2006 (NC3Rs, 2006). Cages were approximately 2.5 M high by 4 M long by 2 M deep, constructed with high level observation balconies and with a floor of deep litter to allow foraging. Additional environmental enrichment was afforded by the provision of toys, swings, feeding puzzles and DVDs for visual stimulation. In addition to standard old-world primate pellets, diet was further supplemented with a selection of fresh vegetables and fruit. For each procedure, sedation was applied by intramuscular injection with ketamine hydrochloride (10 mg/kg) (Ketaset, Fort Dodge Animal Health Ltd., Southampton, UK). None of the animals had been used previously for experimental procedures and each socially compatible group was randomly assigned to a particular study treatment. The study design and all procedures were approved by the UKHSA Porton Down Animal Welfare and Ethical Review Body, and authorised under an appropriate UK Home Office project license.

Animals were sedated at regular intervals to measure body weight and temperature, red blood cell (RBC) haemoglobin levels and erythrocyte sedimentation rate (ESR) and for collection of blood samples, X-radiographs or CT scans. Animal behaviour was observed throughout the studies for contra-indicators and the time of necropsy, if prior to the end of the planned study period, was determined by experienced primatology staff based on a combination of the following adverse indicators: depression or withdrawn behaviour, abnormal respiration (dyspnoea), loss of 20% of peak post-challenge weight, ESR levels elevated above normal (>20 mm), haemoglobin level below normal limits (<100 g/dL), increased temperature (>41 °C) and abnormal thoracic radiograph.

The studies reported describe individuals with active TB that was either controlled during the study period or progressed to levels that met humane endpoint criteria. TB was not latent in any of the subjects.

### 2.2. Experimental Groups

Once CT images analysis was concluded, we defined five experimental groups among the thirty-five macaques, based on the genotype and their vaccination status: cynomolgus unvaccinated (CUV, n = 10), cynomolgus intradermal-BCG-vaccinated (CIDV, n = 3), rhesus unvaccinated (RUV, n = 11), rhesus aerosolized, or inhaled, BCG-vaccinated (RAEV n = 6) and rhesus intradermal-BCG-vaccinated (RIDV, n = 5). These groups together with CT scan schedule are shown in Figure 1 and Appendix A.

### 2.3. CT Images

We analysed 123 computed tomography (CT) scans obtained from subjects enrolled in different studies [13,14,28] where sequential images were acquired from each macaque on three or four occasions during the post-infection period from the 3rd to the 16th week (Figure 1 and Appendix A).

Chest multidetector CT scans were obtained by using a 16 slice Lightspeed CT scanner (General Electric Healthcare, Milwaukee, WI, USA), from sedated subjects while free breathing. Technical parameters applied: tube voltage, 120 kVp; tube current modulation, 100–150 mA. Reconstructions were made using a high spatial frequency algorithm and lung window at a slice thickness of 0.625 mm. These technical parameters are extracted from the original animal experimental studies [13,14,28].

### 2.4. TB Lung Lesions Analysis from CT Images

CT scans were evaluated by a medical consultant radiologist with expertise in respiratory diseases and her supervisor. Both were blinded to vaccination status and clinical data. Data show the consensus between both. Images were then examined in Philips IntelliSpace Portal software (2015 Koninklijke Philips N.V.) by multiplanar reformatting axial, coronal and sagittal, and maximum intensity projection (MIP) to detect the smallest nodules. The Tumor Tracking tool from the mentioned software was used to analyse the volume and main axis of each lesion. We used standard image descriptors, accepted and reproducible concepts from radiologic lexicon detailed on Fleischner Society Glossary as well as in previous studies based on CT images [63].

A.Micronodule is a small solid lung nodule with smooth margins.B.Consolidation (also referred to as consolidated lesion) is a pneumonic patch or necrotizing consolidative process that occupies and even destroys alveoli, and it is described in CT images as a soft tissue lesion within lung parenchyma showing irregular margins.C.Cavitation corresponds to the appearance of gas within lung consolidations. It is described as a central radiolucency image, which matches with gas density, surrounded by soft tissue [8,57].D.Pleural distance is defined as the distance between the margin of each consolidation to the closest pleura or fissure.

### 2.5. Statistical Assessment

To compare the mean and variance of the different observations in the two groups we used the one-way ANOVA test and Kruskal–Wallis test to determine the existence of significant differences. To compare the observed distribution of data into discrete categories, we built a contingency table and the Fisher’s test. Implementation was conducted in MATLAB and Prism. We computed the mean with available data. In the plots, we indicated with a dashed line if there were missing data. Recollected data are available in the Appendix A.

## 3. Results

### 3.1. Preliminary Images’ Assessment and Anatomy

We were provided with high-quality CT scan images, similar to those acquired for medical diagnosis in humans. A first anatomical examination corroborated that cynomolgus macaques had a smaller lung volume than rhesus macaques (Appendix A). Overall, 323 consolidations were identified; 285 were found in rhesus and 38 in cynomolgus, with a median of 14 and 2 lesions per animal, respectively (Table 1). A total of 2228 micronodules were detected, the majority of which (1924) were found in rhesus macaques, while 304 micronodules were detected in cynomolgus (Table 2). Collected data revealed that lesions globally increase over time. Right lung lobes developed more consolidations in number but left lung lobes presented with a higher proportion occupied by disease. A non-significant trend was seen for the major involvement to occur in the upper lobes, which was higher in the left side (Appendix A). Further CT scans’ inspection showed the lung lobes to be separated by pleural fissures as well as bronchovascular bundles, thus presenting the same airway anatomy pattern seen in humans. The right lung is the largest and is composed of upper, middle, lower and infracardiac (or accessory or azygous) lobes, and the left lung is composed of upper, middle (or lingula) and lower lobes [64,65,66,67,68,69]. However, in our study, the fissure for the middle left lobe was frequently incomplete or even absent.

We established the following measurements per each lesion:

For micronodules, we considered all lesions with a maximum axis ranging from 1 to 4.4 mm (which equates to a maximum volume of 0.044 cm^3^). Their length mainly falls in the granuloma’s size (1–3 mm), but we also included rounded spherical lesions with a maximum axis under 4.5 mm, as they tended to be stable. In contrast, almost all lesions above 4.5 mm increased or showed an instable size. We defined two subgroups: (a) Daughter micronodules, found around consolidations, such as satellites, that are presumably generated from that consolidated lesion [5,49,54]. They are located within an annular zone, where the amplitude of this circular crown corresponds to the consolidation’s short axis; they were counted for each CT scan (Figure 2). (b) Isolated micronodules, the ones independent or not associated with a consolidated lesion (Figure 3). The size and number of them were determined together within their bronchopulmonary segment.

Consolidations are large enough to be followed up and carefully tracked over consecutive scans, and their axis (mm) and volume (cm^3^) were measured (Figure 4). We considered that these kinds of lesions modify their size over time and are equal or bigger than 4.5 mm.

Cavitations must have a greater size than the surrounding bronchial calibre, for avoiding misinterpretation with air bronchogram. It was assessed qualitatively, being present (yes) or absent (no).

In pleural distance measurements we also determined the lung volume from each animal, using the COPD tool from the Philips IntelliSpace Portal software, which allowed the evaluation of the main airway and volume from both lungs separately.

We also observed foci of bronchocentric pneumonia, detected as a ‘tree-in-bud’ pattern in CT images, which is a common CT finding in active lung TB [58,59,60,61,70,71]. However, we did not include them in our analyses due to their lack of homogeneity and instability, which made them difficult to demarcate and to follow-up.

RUV05 presented with miliary tuberculous disease and developed hundreds of lung micronodules distributed diffusely and homogeneously (as well as extrapulmonary miliary disease). Such profuse dissemination is well known and has been described in other studies as ‘widespread discrete pulmonary nodules’ [63,72]. It is uncommon and denotes extremely uncontrolled infection with hematogenous dissemination. Our study was focused on pulmonary pneumonia and airway dissemination; hence, we removed the data collected from this animal from the statistical evaluation due to the different TB disease.

### 3.2. Mtb-Induced Disease Is Worse in Rhesus Macaques

In order to exemplify the complexity of the operational procedure, Figure 5 illustrates a particular evolution in one case (RIDV02). Overall development of the natural infection in control groups is summarized in Figure 6. The temporal evolution of the lesions is shown per animal, based on the size and localization per lobe in a 2D representation. Each pair of lungs is reshaped according to each individual volume (adapted to the image obtained from macaque RUV04).

At first sight, this summary highlights the greater severity of the infection in rhesus. In particular, at week 3, cynomolgus developed a low number of lesions, with a small volume, which were usually well controlled and exhibited minimal enlargement with time. Lesions were not identified in one individual (CUV10), although infection was confirmed by other methodologies. Other individuals showed a decline in condition only at the late phase (15th week) of the study, e.g., CUV07, in which an extraordinary growth of consolidations was detected on CT images that was probably due to total main bronchus stenosis (despite it being uncommon in cynomolgus).

In contrast, lesions in rhesus were abundant and larger at week 3, usually exhibiting enlargement and dissemination over time. Poor control leading to extreme disease was evident in RUV05 (which developed profuse miliary dissemination, described above) and also in RUV06 and RUV08; these animals had to be euthanized before finishing the monitoring period due to progressive disease that met humane endpoint criteria. However, there was at least one case in a rhesus where a spontaneous weak regression of lesions was observed (RUV10) (Figure 6).

Cynomolgus macaques developed significantly fewer lung consolidations and micronodules, resulting in a smaller pulmonary volume occupation than that seen in rhesus macaques, as shown in Table 1 and Figure 7. Cynomolgus also had a significantly lower percentage of daughter micronodules than rhesus (Table 1 and Table 2, and Figure 8). The overall ratio of daughter/isolated micronodules for non-vaccinated macaques comprising all time points was 1.32 and 0.34 for rhesus and cynomolgus, respectively. Similarly, the number of consolidated lesions associated with daughter micronodules was also significantly lower in cynomolgus (Table 3). These data reflect a higher containment of lesions’ progression in cynomolgus.

The size of consolidated lesions was highly dependent on the number of daughter micronodules. This is reflected in Figure 9 where a clear and significant correlation is noted between the size of the consolidated lesions and the number of daughter micronodules, regardless of the vaccination status. This positive slope is markedly higher in rhesus. In addition, a greater portion of occupied pulmonary volume is correlated to a higher number of isolated micronodules (Appendix A), so the higher the infectious involvement of lungs, the higher the rate of pulmonary dissemination (local and endobronchial reinfections, which fits with the Dynamic Hypothesis [52]).

We also detected some transitional lesions, which correspond to isolated micronodules that increased to a size of more than 4.4 mm from one CT to the next, thus becoming a consolidation (Table 1). This enlargement occurred mainly in the early phase between the third and eighth week post-challenge. Remarkably, rhesus developed directly a large number of consolidations in the early phase after infection, in contrast to cynomolgus; hence, transitional lesions represent a higher impact on cynomolgus. Importantly, there is no significant difference between both macaque genotypes considering the progression of isolated micronodules towards consolidated lesions.

### 3.3. BCG Vaccination in Rhesus Macaques Reduces the Number of Lesions, Although Intradermal Vaccination Better Controls Daughter Micronodules

Initial evaluation of the impact of vaccination in macaques suggested that a protective effect appeared by the 3rd week post-challenge (Figure 6, Figure 8 and Figure 10). However, a deeper analysis revealed that protection measured in terms of infection (i.e., occupied lung fraction at week 3), did not appear to provide a reliable indicator of later outcome (Appendix A). When each disease volume fraction was normalized against the maximum infiltrated lung volume per macaque (Figure 11), it revealed a clear impact of the vaccination in cynomolgus macaques early after challenge, although it did not last until the end of the monitoring time.

Due to the correlation between the volume of the consolidated lesions and the number of daughter micronodules, we investigated if vaccination had an impact on this fraction of micronodules in relation to the enlargement of consolidated lesions, and this was the case. Looking at Figure 8, vaccination clearly reduced this fraction in all vaccinated macaques, although temporally in rhesus, as it rapidly resumed by week 7 to 8. The reduction in this parameter only prevails in cynomolgus until the 15th week.

Evaluation of the route of vaccination in rhesus revealed an interesting paradox. While aerosol vaccination (RAEV) reduced the total number of micronodules, more than intradermal vaccination (RIDV), a significantly greater fraction of daughter micronodules were present (3.03) (Table 4). Thus, the protective effect of RAEV relies on the reduction in the total number of micronodules (Table 2 and Table 3) rather than the control of subsequent consolidation development. Even though the reduction in total micronodules was weaker in the RIDV, its effect was noted in both types of micronodules, even when this fraction was slightly higher in RIDV (1.51) than non-vaccinated (RUV) (1.32) (Table 2). At the end, this means that progression towards consolidation at a hypothetical later time point would be faster in the RAEV group, followed by RIDV and RUV.

The effect of vaccination in rhesus was also observed as a decrease in the proportion of consolidated lesions that were associated with daughter micronodules, which was more pronounced and significant after intradermal (95%) than after aerosol vaccination (99%) (Table 3 and Table 5). This effect was even more pronounced in cynomolgus, although in this case there was no variation in the proportion of isolated and daughter micronodules between vaccinated (intradermal) and unvaccinated groups (Table 2 and Table 4). This can be explained because, as indicated above, cynomolgus are able to spontaneously stop the progression of the consolidated lesions soon after the infection (i.e., before week 3 after challenge).

### 3.4. Cavitation Is More Probable in Larger Lesions and Contact with Pleura Appear to Impact on TB Progression

Cavitation occurred mostly in rhesus and especially at the late phase, where 34.4% of consolidated lesions developed cavities in non-vaccinated rhesus (RUV), closely followed by 28.4% of the aerosolized (RAEV) and by 24.3% of the ID (RIDV). Analysis of cavitation in rhesus (Figure 12) revealed a relationship with lesion size (*p* < 0.0001): the higher the consolidation volume, the higher the rate of cavitation, with cavitation more probable in lesions greater than 2 cm^3^, which represents a rate of 57%. In contrast, almost 80% of lesions from 0 to 2 cm^3^ did not cavitate. The anatomic distribution of lesions was not found to influence cavity formation.

We also evaluated the relationship between consolidations and pleura, in order to investigate if the contact with pleura favoured the encapsulation process of the lesions and stopped enlargement. As shown in Appendix A, the group of smaller lesions (up to 0.5 cm^3^, which accounts for the higher number of consolidations) were mainly not in contact with the pleura, especially in rhesus (which have bigger lungs), and from this size on, consolidations were mainly in contact with the pleura.

The next step was to determine if consolidations changed between two consecutive CT scans, with respect to their size and distance from pleura. Thus, we evaluated the change in the volume together with the relationship with the contact with pleura. We explored the change in the relative volume gradient, explained through a boxplot (Figure 13). In this regard, differences are significant (*p* = 0.04 Kruskal–Wallis) in cynomolgus for those consolidations that make contact with the pleura, as they remain stable in volume. Thus, pleural contact is protective in cynomolgus, preventing the progression of consolidations, and this trend is irrespective of their vaccination status. In contrast, in rhesus, consolidations that enlarge and make contact with pleura do not experience a volume control because the increase in the size is too fast and large to benefit from the encapsulation process activated by the pleura.

## 4. Discussion

TB remains a major killer of humankind, and the lack of a surrogate of protection, means there is a constant need for the field to find experimental models able to elucidate new biomarkers and correlates to assist the development of new more effective vaccines. Nowadays, the macaque is considered the most clinically relevant experimental infection model of TB given its physiological closeness with humans and the similarity in disease manifestations. However, the macaque does lack the interlobular septa, present in human lungs, which have been demonstrated to provide a protective mechanism in the minipig model [9]. The outcome of experimental infection with Mtb has been widely studied in macaques, and models established in rhesus and cynomolgus have been used to assess the efficacy of new drugs and vaccines. Mtb infection in cynomolgus resembles what occurs in humans, with the display of the whole spectrum of human TB, mainly controlling the infection and becoming latently infected. In contrast, rhesus experience a rapid dissemination of the infection in the lung, with the development of active TB in almost all the infected animals, a characteristic that enables the capacity of vaccines to protect against this progression to be assessed using manageable group sizes [73].

There are little data on vaccines showing significant protection against Mtb challenge in the macaque model with reports limited to BCG delivered intravenously (IV), for which the clinical deplorability is debatable, or a CMV-vectored vaccine [74] that is at the preclinical stage of development, and MTBVAC [28]. While there are very little data on immune responses induced in NHP and humans after exactly the same immunizations, the immune profiles induced following immunization with MTBVAC were shown to reflect those identified in human clinical trials [75,76,77]. This concordance between immune profiles measured in clinical trials and a macaque preclinical assessment of a novel TB vaccine candidate, demonstrating significantly improved outcomes after Mtb challenge, is a promising indication that the protection provided by MTBVAC vaccination to macaques will translate to the human population.

The ability to demonstrate vaccine efficacy is critical, and the role of the macaque models in vaccine development is underpinned by the ability to measure vaccine-associated impacts on the disease that develops following infectious challenge.

Our approach has been to test a new way to interpret data derived from these models based on the ‘Bubble Model’, that defines the progression from latent to active TB supported by the interpretation of disease progression defined in the studies on the C3HeB/FeJ mice [49,54,78,79]. As with humans, the C3HeB/FeJ mouse model establishes that the development of a large lesion after the Mtb infection requires a strong inflammatory response in the initial lesions, the induction of daughter micronodules around them and the coalescence of all lesions. To evaluate this concept in macaques, we used CT images obtained previously from studies using an experimental model of low-dose aerosol exposure, thus closely resembling what is thought to happen in the natural infectious process, where CT scans were collected at several time points to monitor the evolution of the infection in each animal [13,14,28]. In line with previous reports, three weeks after infection, more large lesions were identified in rhesus compared to cynomolgus, which has been related to the development of a stronger inflammatory response in the former [14,40]. These lead to an exudative lesion, a hallmark of progression towards active TB [51] and a key issue for its development, according to the bubble model.

Thus far, the assessment of new vaccines has been mainly based on the induction of an immune response and a reduction in disease burden measured using pathology scoring, which included a semiquantitative approach of the involvement of lungs and other organs, as well as imaging-based measures through PET-CT [11,45], CT evaluation [28,80], histology-based measures (e.g., granuloma staging and prevalence), together with bacterial burden in organs and changes in clinical parameters. There is quite a lot of experience in using imaging as a tool to evaluate BCG efficacy in animal models, with special interest in NPH. Some studies appeared during the last few years that mainly focused on BCG alternative routes. White et al. [28] analysed CT images in rhesus and used a quantitative score system for the estimation of TB-induced disease burden [63]; Sharpe et al. [81] and recently Sibley et al. [29] used MRI images in ex vivo lungs from macaques to determine pulmonary disease burden applying stereology [46], as well as counting discrete and coalescent lesions; Darrah et al. [72] and DiFazio et al. [82] performed PET-CT images in macaques and mainly evaluated the grade of tuberculous activity by measuring global lung parenchyma FDG avidity, as well as counting lesions (which were called granulomas). Moreover, there are some experiences including other animal models, as Kraft et al. [83] analysed ex vivo lungs from guinea pigs using MRI, and described their imaging appearance, their anatomical distribution and nodule counting.

Previous analysis of these images highlighted the early reduction in micronodules after BGC vaccination [28]. Our work was able to advance deeper in the exploitation of these data by differentiating the nodules related with the consolidated lesions, daughter micronodules, from the isolated ones. According to the ‘Bubble model’, daughter micronodules emerge from an initial isolated micronodule, increase in size, consolidate creating larger nodules, which in turn develop more daughter micronodules, a process that enlarges the lesion to the point that it becomes visible by chest X-ray, the hallmark for classifying an Mtb infection as active TB [51]. Data from the C3HeB/FeJ model suggests this progression mainly develops at the beginning of the infection when there is still a lower impact of the adaptive immune response. This is characterized by the infiltration of neutrophiles, which allows the extracellular growth of Mtb, fuelling enlargement and allowing the spread of close new infectious foci due to the drainage of infected foamy macrophages, which in turn attracts new neutrophils, causing the development of daughter lesions [54]. This led us to draw a parallel with the NHP model. On the one hand, rhesus develop highly inflammatory lesions with a high neutrophilic infiltration, a process that might be related with the progression associated to the development of daughter micronodules. Conversely, cynomolgus develop less inflammatory and more contained lesions [40], which would link to a reduced capacity to generate daughter micronodules. The lower inflammatory response, and thus the lower capacity to generate daughter cells, allows the fibroblasts to encapsulate the lesion, stopping its progression [9], and explains why the vicinity of the pleura, and its encapsulation capacity, has only a protective effect in cynomolgus.

Our data provide two new measures of pulmonary disease burden induced following infection with Mtb, namely, the ratio daughter/isolated micronodules and the percentage of consolidated lesions linked to daughter micronodules identified through CT in macaques. These measures provided a new tool to evaluate the protective effect of the licenced vaccine (BCG) and revealed differences in disease progression following either intradermal or mucosal (inhaled) administration, even when both routes were able to reduce the number of consolidated lesions and generated a similar reduction in the number of micronodules. While this mucosal aerosol vaccination was better able to reduce the number of isolated micronodules, potentially because of a better capacity to reach and stimulate the immune response in the whole lung, intradermal administration should have an improved ability to reduce the emergence of daughter micronodules, which are the ones responsible for the progression towards active TB.

In summary, we defined a new tool for the evaluation of pulmonary disease induced by Mtb infection, through the description of ‘daughter micronodules’, which provides a better understanding of the progression of TB and which can be applied for testing new vaccines. This is also amenable for development as a machine learning tool, as nodules are easily identifiable in CT images and can be quantified. Enhanced measures of TB disease burden and progression, such as those reported here, will refine the models used in the development of new vaccines and reduce the number of animals required, consequently providing important scientific and welfare benefits. Equally, the use of data across studies increased the power of the analysis applied and obtained more information without requiring the use of further macaques in line with the principles of the 3Rs.

## Figures and Tables

**Figure 1 pathogens-12-00236-f001:**
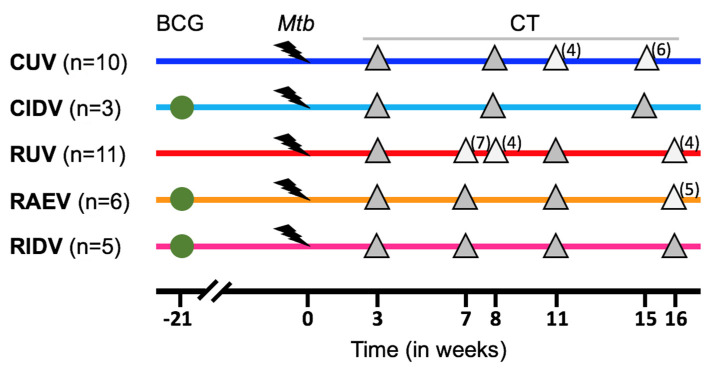
**Study schedule.** Macaques received BCG vaccination (green circles) delivered by aerosol or intradermal injection, or were left as unvaccinated controls. All animals received ultra-low dose aerosol challenge with Mtb 21 weeks after. Gray triangles indicate application of ‘in vivo’ CT scanning. Light grey triangles indicate the time points where not all animals were scanned and the number between brackets shows in how many of them this was effectively performed. In blue, unvaccinated cynomolgus (**CUV**). In cyan, intradermal-vaccinated cynomolgus (**CIDV**). In red, unvaccinated rhesus (**RUV**). In orange, aerosol-vaccinated rhesus (**RAEV**). In purple, intradermal-vaccinated rhesus (**RIDV**).

**Figure 2 pathogens-12-00236-f002:**
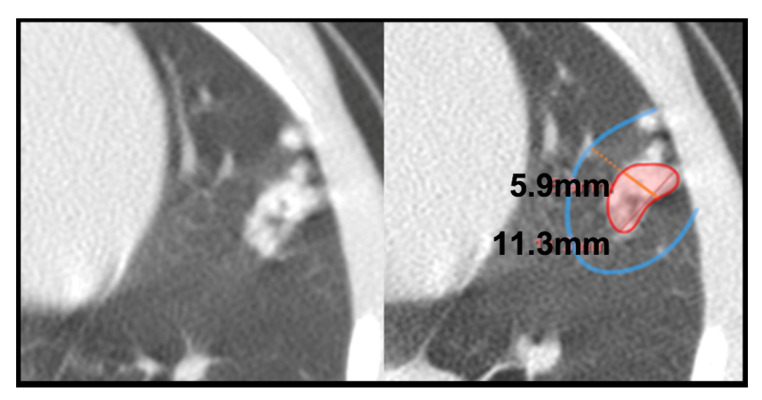
**Consolidation with daughter micronodules**. Left picture shows original CT image from a lesion located in the inferior lingular bronchopulmonary segment at 11th week scan in an unvaccinated rhesus (RUV10). Right picture exposes that daughter micronodules are those situated within a surrounding area that has the same length as the consolidation’s short axis. At same time, this figure exhibits the lesional complex consisting of a consolidation with daughter micronodules.

**Figure 3 pathogens-12-00236-f003:**
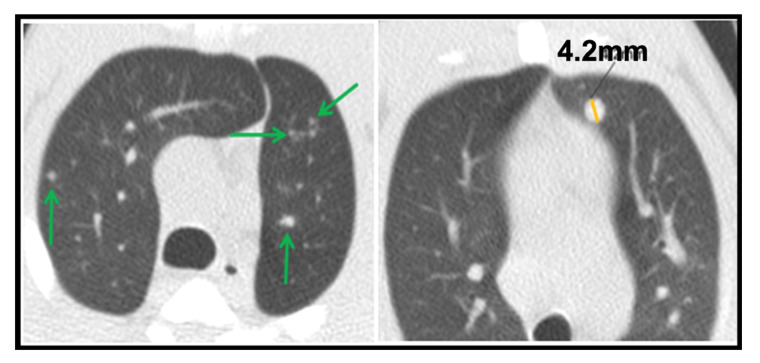
**Isolated micronodules.** Left picture shows several small isolated micronodules distributed on both upper lung lobes in an intradermic-vaccinated rhesus (RIDV02) at 3rd week scan. Right picture exhibits a large isolated micronodule (4.2 mm) located in the anterior bronchopulmonary segment in the left upper lobe at 3rd week scan in an intradermic-vaccinated rhesus (RIDV04). Notice that nodular images not labelled correspond to pulmonary vessels.

**Figure 4 pathogens-12-00236-f004:**
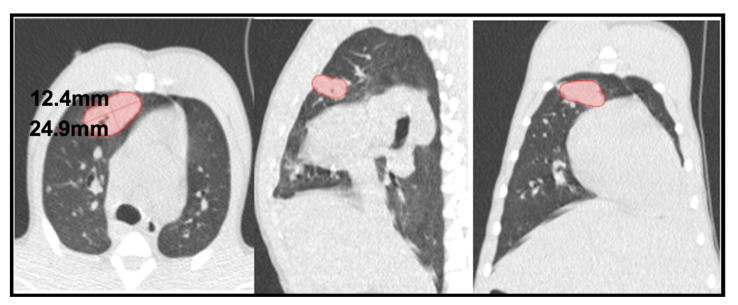
**CT tridimensional evaluation**. Example of a cavitated consolidation in an intradermic-vaccinated rhesus (RIDV02) at 16th week (pictures from left to right: axial, sagittal and coronal CT scan planes). Its main axis measures 31.4 mm and its volume reaches 2.308 cm^3^.

**Figure 5 pathogens-12-00236-f005:**
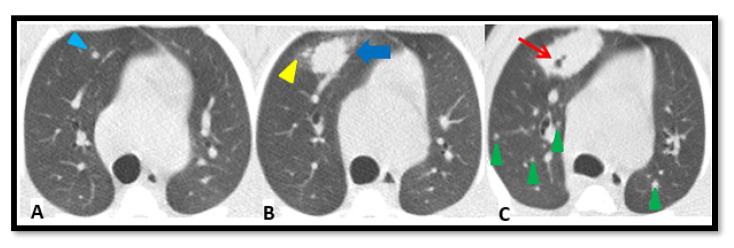
**Pulmonary TB progression**. Lesions from 3rd to 16th post-challenge week in an intradermic-vaccinated rhesus (RIDV02). (**A**): A 3rd week CT shows a lesion in the anterior bronchopulmonary segment in the right upper lobe that initially is a 2.1 mm isolated micronodule (light blue arrowhead). (**B**): An 11th week scan exhibits its progression to a consolidation (dark blue arrow) and the development of two daughter micronodules (yellow arrowhead). (**C**): CT performed at 16th week shows a higher enlargement of the consolidation, in which arose a cavitation (red arrow). Moreover, some small isolated micronodules (green arrowheads) appeared distributed on both lungs. Notice that nodular images not labelled correspond to pulmonary vessels.

**Figure 6 pathogens-12-00236-f006:**
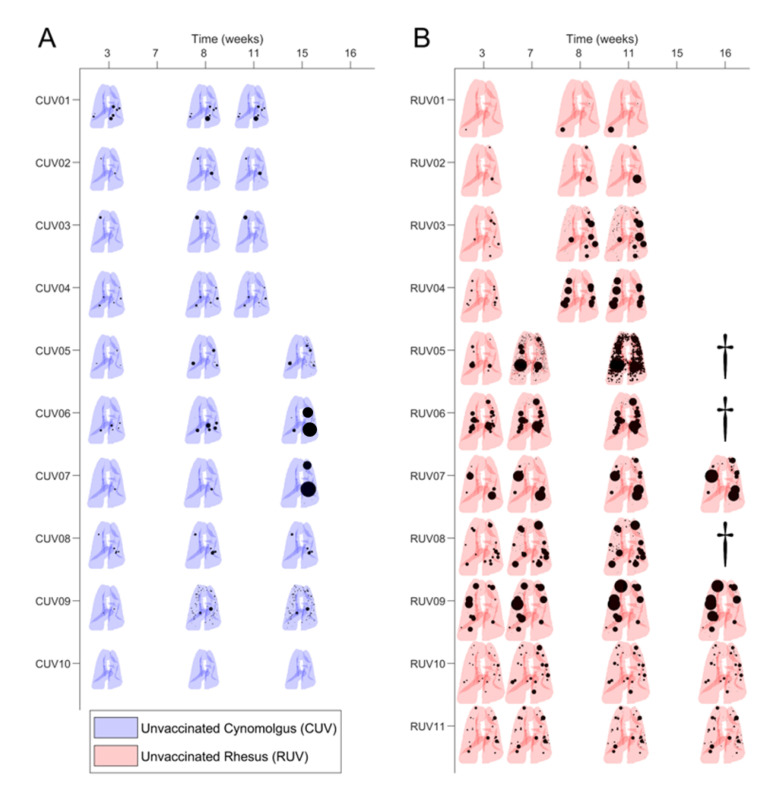
**Evolution of lesions in unvaccinated macaques.** The size and location of TB lesions determined from computed tomography (CT) scan taken at specific times after initial infection are shown in each pair of lungs. Lesions (consolidations and micronodules), in black, are represented as spheres of the measured volume and located in the segment where they are identified. (**A**) In blue, reconstructions of CT scans from each unvaccinated cynomolgus. (**B**) In red, reconstructions of CT scan collected from each unvaccinated rhesus. Macaques RUV05, RUV06 and RUV08 marked with dagger symbol were euthanized before the end of the study.

**Figure 7 pathogens-12-00236-f007:**
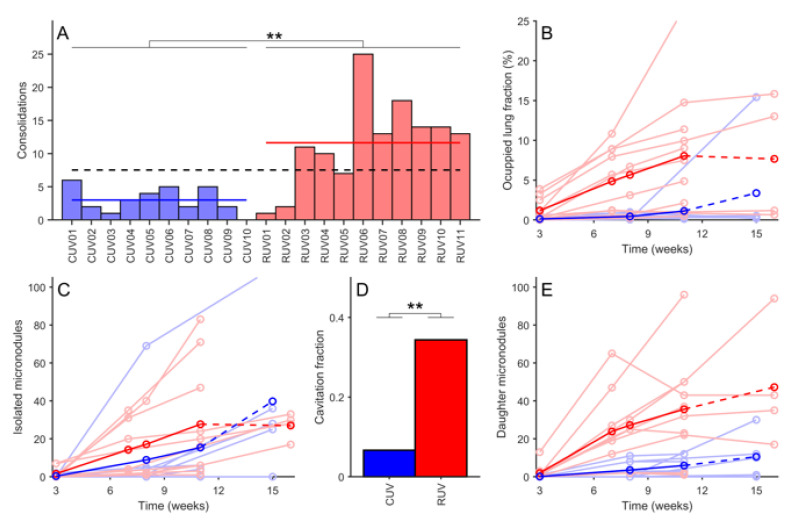
**Analysis of unvaccinated macaques.** (**A**) displays the number of identified consolidations for each macaque at the end of the study. In light blue, each non-vaccinated cynomolgus (CUV) macaque is represented. In blue, the mean number of lesions for CUV macaques. In light red, each non-vaccinated rhesus (RUV) macaque is represented. In red, the mean number for RUV macaques. Dotted black line shows the mean value for all non-vaccinated macaques. (**B**) shows the occupied lung fraction for each macaque. (**C**) shows the number of isolated micronodules for each macaque. (**D**) shows the fraction of consolidations that present cavitation in any CT scan. **E** shows the number of daughter micronodules for each macaque. In (**B**–**E**) each circle represents an experimental measurement from computed tomography (CT) scans. Light lines represent the evolution for each macaque. Darker lines are the mean lines’ value. Dotted lines are used when one or more macaques are not considered because they were euthanized, or the study ended before that time point. Data is analysed through one-way ANOVA test (** *p* < 0.01).

**Figure 8 pathogens-12-00236-f008:**
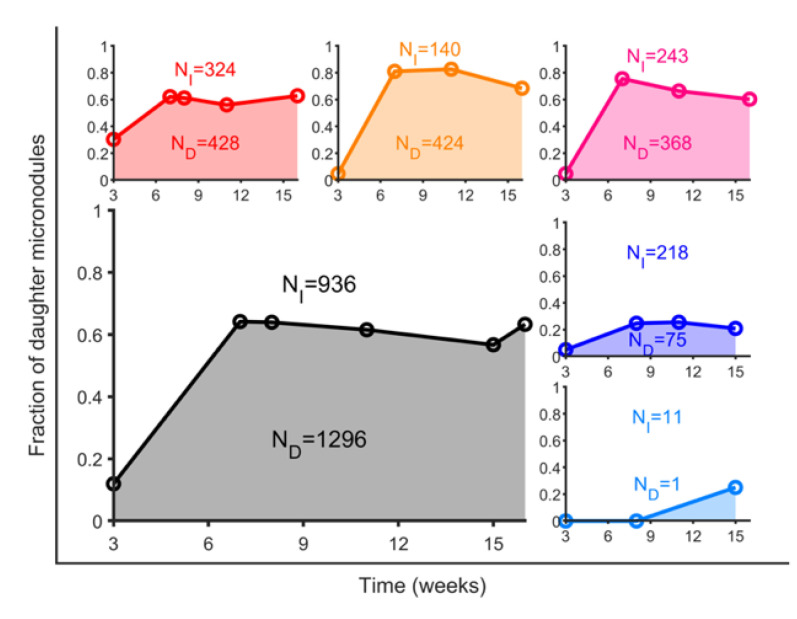
**Fraction of daughter micronodules**. Number of total isolated micronodules (N_I_) and number of total daughter micronodules (N_D_) are shown in grey. Total number of micronodules is computed considering all CT scans independently. Total fractions are depicted in blue for unvaccinated cynomolgus (CUV); in cyan, for intradermal-vaccinated cynomolgus (CIDV); in red, for unvaccinated rhesus (RUV); in orange, for aerosol-vaccinated rhesus (RAEV); and in purple, for intradermal-vaccinated rhesus (RIDV).

**Figure 9 pathogens-12-00236-f009:**
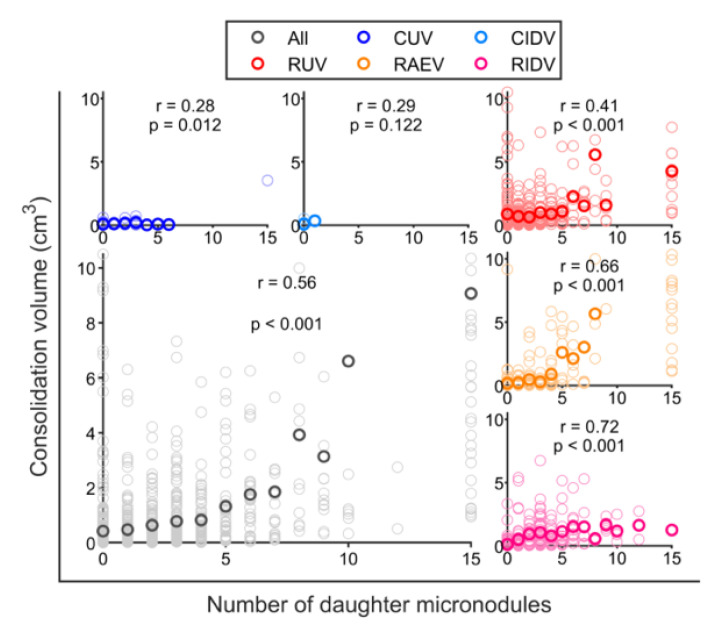
**Correlation between the volume of consolidated lesions and the number of daughter micronodules**. Light circles represent a single consolidation. Dark circles represent the mean volume value of all consolidations that present the same number of daughter micronodules. Correlation value between both quantities is written as *r*. The significance of correlation value is determined from the *p*-value that is shown under correlation value, according to the X test. In blue, unvaccinated cynomolgus (CUV); in cyan, intradermal-vaccinated cynomolgus (CIDV); in red, unvaccinated rhesus (RUV); in orange, aerosol-vaccinated rhesus (RAEV); in purple, intradermal-vaccinated rhesus (RIDV); in grey, all macaques (all).

**Figure 10 pathogens-12-00236-f010:**
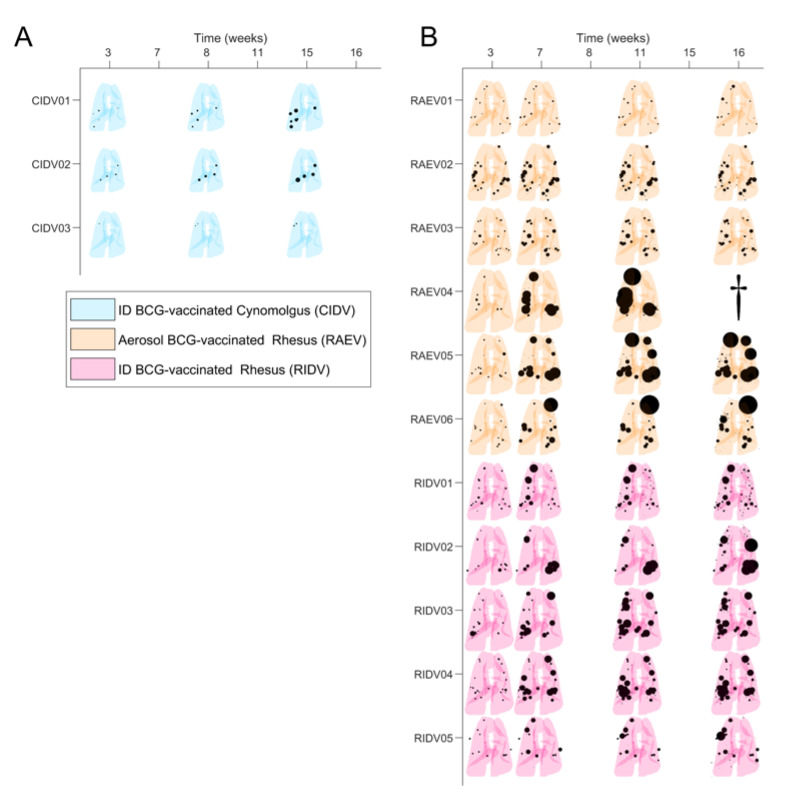
**Lesions’ evolution in vaccinated macaques.** Each pair of lungs show the size and location of TB lesions determined from computed tomography (CT) scan taken after initial infection. Lesions (consolidations and micronodules), in black, are represented as spheres of the measured volume and located in the segment where they were identified. (**A**) In cyan, the reconstructions of lungs from intradermal-vaccinated cynomolgus (CIDV) for each CT scan. (**B**) In orange, the reconstructions of lungs from aerosol-vaccinated rhesus (RAEV) for each CT scan. In purple, the reconstructions of lungs from intradermal-vaccinated rhesus (RIDV) for each CT scan. Macaque RAEV04 marked with dagger symbol was euthanized before the planned end of study, according to the welfare monitoring plan.

**Figure 11 pathogens-12-00236-f011:**
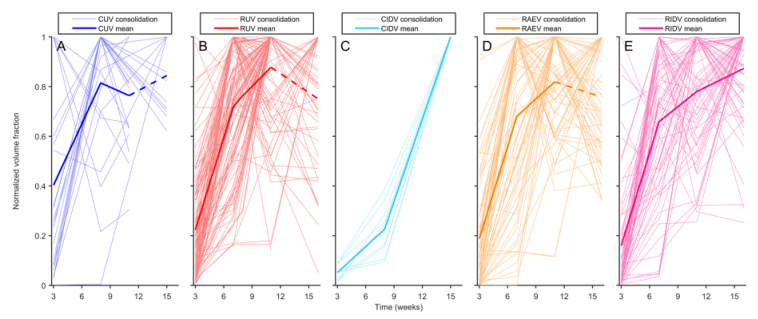
**Evolution of normalized occupied lung volume per each individual and experimental group.** Volume of each involved area is divided by the maximum occupied lung volume reached by each individual. (**A**) Light blue lines indicate values for unvaccinated cynomolgus (CUV). Blue line indicates the mean value of all light blue lines. (**B**) shows light red lines for unvaccinated rhesus (RUV). Red line is the mean value of all light red lines. (**C**) shows light cyan lines for intradermal-vaccinated cynomolgus (CIDV). Cyan line shows the mean value of all light blue lines. (**D**) shows light orange lines are for aerosol-vaccinated rhesus (RAEV). Orange line is the mean value of all light blue lines. (**E**) shows light purple lines are for intradermal-vaccinated rhesus (RIDV). Purple line is the mean value of all light blue lines. Dotted line appears for some lesions where data are not available due to study finalization or euthanized macaque.

**Figure 12 pathogens-12-00236-f012:**
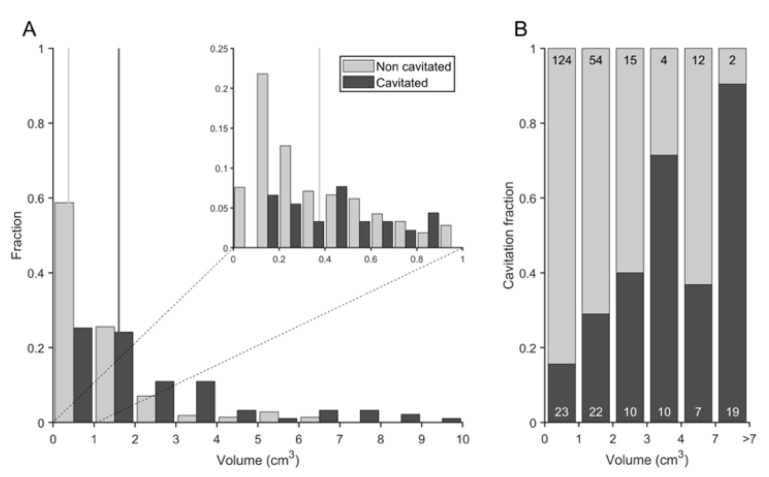
**Relation between the fraction and the volume of cavitated consolidations in rhesus macaques**. (**A**) Consolidation volume distribution for non-cavitated (light grey) and cavitated (dark grey) lesions. Vertical lines show the median values for each of the subsets. (**B**) Cavitation fraction for the consolidations that are in a desired range of volumes: 0 to 1 cm^3^, 1 to 2 cm^3^, 3 to 4 cm^3^, 4 to 7 cm^3^ and consolidations bigger than 7 cm^3^. The number of non-cavitated consolidations observed is shown in black, at the top. In white, at the bottom, the number of cavitated consolidations observed.

**Figure 13 pathogens-12-00236-f013:**
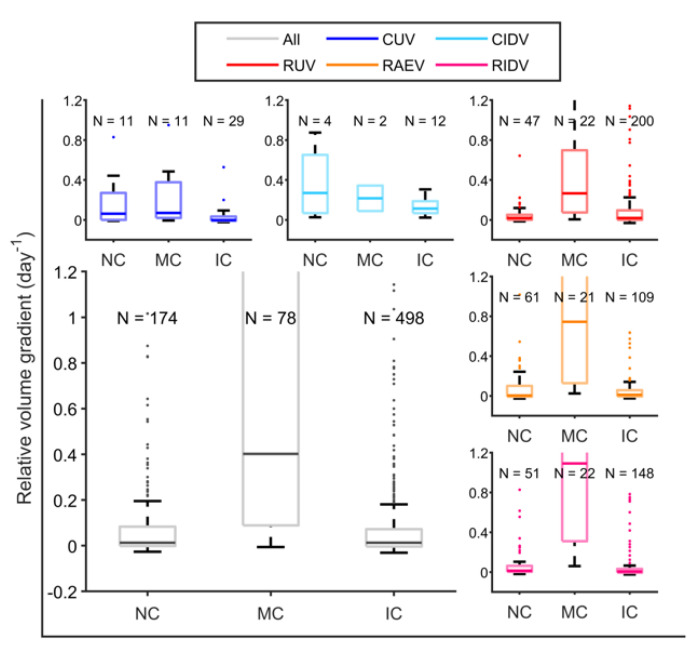
**Volume increase in the consolidations in relation with the contract with pleura**. The increase in volume is expressed as relative gradient per day between two consecutive CT scans. Consolidations are classified between: not in contact (NC), if in both CT scans it is not in contact with pleura; make contact (MC), if it is not in contact in the first CT but it is in contact in the second; and in contact (IC), if in both CT scans it is in contact with pleura. It is indicated how many cases are identified for each classification. In blue, unvaccinated cynomolgus (CUV); in cyan, intradermal-vaccinated cynomolgus (CIDV); in red, unvaccinated rhesus (RUV); in orange, aerosol-vaccinated rhesus (RAEV); in purple, intradermal-vaccinated rhesus (RIDV); in grey, all macaques (all).

**Table 1 pathogens-12-00236-t001:** Distribution of the lesions, comprising all time points and experimental groups.

EG	Consolidations			Daughter Micronodules			Isolated Micronodules		
	Median	Range	%	Median	Range	%	Median	Range	%
**CUV (n = 320)**	2	[0 6]	9	5.5	[0 30]	23	5.5	[0 110]	68
**CIDV (n = 22)**	4	[0 6]	45	0	[0 1]	5	4	[2 5]	50
**C (n = 342)**	**2**	**[0 6]**	**11**	**3**	**[0 30]**	**22**	**4**	**[0 110]**	**67**
**RUV (n = 871)**	12.5	[1 25]	14	36	[2 96]	49	30	[1 83]	37
**RAEV (n = 643)**	14.5	[0 19]	12	54	[0 151]	66	18.5	[14 37]	22
**RIDV (n = 697)**	15	[12 24]	12	55	[45 138]	53	29	[21 117]	35
**R (n = 2211)**	**14**	**[0 25]**	**13**	**51**	**[0 151]**	**55**	**29**	**[1 117]**	**32**
**TOTAL (n = 2553)**	**10.5**	**[0 25]**	**13**	**27.5**	**[0 151]**	**51**	**20**	**[0 117]**	**37**

**EG:** Experimental group. **CUV**: unvaccinated cynomolgus; **CIDV**: intradermal-BCG-vaccinated cynomolgus; **C**: total cynomolgus; **RUV**: unvaccinated rhesus; **RAEV**: aerosol-BCG-vaccinated rhesus; **RIDV**: intradermal-BCG-vaccinated rhesus; **R**: total rhesus. **n** means the total number of lesions per each experimental group. Median shows the value separating the higher half from the lower half of the number of each type of lesion per experimental group.

**Table 2 pathogens-12-00236-t002:** Distribution of micronodules, comprising all time points and experimental groups.

EG	Number (n)			Fraction DM/IM
	Total	DM	IM	
**CUV**	293	75	218	0.34
**CIDV**	12	1	11	0.09
**C**	**304**	**75**	**229**	**0.33**
**RUV**	752	428	324	1.32
**RAEV**	564	424	140	3.03
**RIDV**	611	368	243	1.51
**R**	**1924**	**1216**	**708**	**1.72**

**DM:** Daughter micronodules; **IM**: isolated micronodules; **EG:** experimental group. **CUV**: unvaccinated cynomolgus; **CIDV**: intradermal-BCG-vaccinated cynomolgus; **C**: total cynomolgus; **RUV**: unvaccinated rhesus; **RAEV**: aerosol-BCG-vaccinated rhesus; **RIDV**: intradermal-BCG-vaccinated rhesus; **R**: total rhesus.

**Table 3 pathogens-12-00236-t003:** Distribution of consolidations associated with daughter micronodules, comprising all time points and experimental groups.

EG	Consolidations with DM				DM per Consolidation	
	n	%	Median	Range	Median	Range
**CUV (n = 29)**	19	66	2	[0 4]	3	[1 15]
**CIDV (n = 10)**	1	10	0	[0 1]	1	[1 1]
**C (n = 39)**	**20**	**51**	**1**	**[0 4]**	**3**	**[1 15]**
**RUV (n = 121)**	111	92	11.5	[1 24]	3	[1 15]
**RAEV (n = 80)**	79	99	14.5	[0 19]	3.5	[1 15]
**RIDV (n = 88)**	84	95	14	[11 24]	3	[1 15]
**R (n = 289)**	**274**	**95**	**14**	**[0 24]**	**3**	**[1 15]**
**Total (n = 328)**	**294**	**90**	**9**	**[0 24]**	**3**	**[1 15]**

**DM**: Daughter micronodules. **EG:** Experimental group. **CUV**: unvaccinated cynomolgus; **CIDV**: intradermal-BCG-vaccinated cynomolgus; **C**: total cynomolgus; **RUV**: unvaccinated rhesus; **RAEV**: aerosol-BCG-vaccinated rhesus; **RIDV**: intradermal-BCG-vaccinated rhesus; **R**: total rhesus. **n** mean number of lesions.

**Table 4 pathogens-12-00236-t004:** Differences in the proportion of daughter and isolated micronodules per experimental group. Contingency table.

EG	CUV	CIDV	C	RUV	RAEV	RIDV	R
**CUV**	ND						
**CIDV**	0.3057	ND					
**C**	ND	ND	ND				
**RUV**	ND	ND	ND	ND			
**RAEV**	ND	ND	ND	<0.0001	ND		
**RIDV**	ND	ND	ND	0.32431	<0.0001	ND	
**R**	ND	ND	<0.0001	ND	ND	ND	ND

**EG**: Experimental group. **CUV**: unvaccinated cynomolgus; **CIDV**: intradermal-BCG-vaccinated cynomolgus; **C:** total cynomolgus; **RUV**: unvaccinated rhesus; **RAEV**: aerosol-BCG-vaccinated rhesus; **RIDV**: intradermal-BCG-vaccinated rhesus; **R**: total rhesus. Numbers correspond to *p*-values (Fisher’s test).

**Table 5 pathogens-12-00236-t005:** Differences in the proportion of consolidated lesions associated with daughter micronodules per experimental group. Contingency table.

EG	CUV	CIDV	C	RUV	RAEV	RIDV	R
**CUV**	ND						
**CIDV**	0.0014	ND					
**C**	ND	ND	ND				
**RUV**	ND	ND	ND	ND			
**RAEV**	ND	ND	ND	0.0585	ND		
**RIDV**	ND	ND	ND	0.0355	>0.999	ND	
**R**	ND	ND	<0.0001	ND	ND	ND	ND

**EG**: Experimental group. **CUV**: unvaccinated cynomolgus; **CIDV**: intradermal-BCG-vaccinated cynomolgus; **C:** total cynomolgus; **RUV**: unvaccinated rhesus; **RAEV**: aerosol-BCG-vaccinated rhesus; **RIDV**: intradermal-BCG-vaccinated rhesus; **R**: total rhesus. Numbers correspond to *p*-values (Fisher’s test).

## Data Availability

Raw data will be available as Appendix A.

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
