# Peer review of "Surveillance of Daughter Micronodule Formation Is a Key Factor for Vaccine Evaluation Using Experimental Infection Models of Tuberculosis in Macaques"

_pathogens, 2023, doi:10.3390/pathogens12020236_

Round 1

Reviewer 1 Report

The study aimed to evaluate the progression of Mycobacterium tuberculosis infection in experimental non-human primates (cynomolgus and rhesus macaques) models through the spatial monitoring of the micronodules and the identification of consolidations with daughter micronodules. Their work provides two new measures of pulmonary disease burden, the ratio daughter/isolated micronodules and the percentage of consolidated lesion linked to daughter micronodules. These measures provide a new tool to evaluate the protective effect of the vaccine (BCG) and revealed differences in disease progression.

Here are comments for improvement that the Authors need to consider:

Q) In Figure legends, Instead of writing Picture A ; the author can write A only and when the author is referring to it they can refer it as Fig XA where X is a number ( eg 4A, 6A, 9A, etc)

Q) In result section lines 251-253 “A total of 2228 micronodules were detected, the majority of which (1924) were found in rhesus macaques, while 304 micronodules were detected in cynomolgus (Table 2).” But table 2 suggest 305 micronodule in cynomolgus (CUV+CIDV ) and (RUV+RAEV+RIDV) 1927 in rhesus . Can the author clarify?

Q) Line 393 Figure 10 is missing

Q) In the Discussion author has given a general statement that “ Whilst this mucosal aerosol vaccination was better able to reduce the number of isolated micronodules, potentially because of a better capacity to reach and stimulate the immune response in the whole lung “ Although, for rhesus macaques author has taken both conditions (aerosol BCG-vaccinated; intradermal BCG-vaccinated) but not for cynomolgus. 

Q) Why the author has not included the group “ cynomolgus with aerosol BCG vaccinated”?

Q) I table 3 Under the EG category, What author mean by n?

Q) What advantages and drawbacks of using these measures as a new tools to evaluate disease progression? Include it in the discussion

Author Response

  1. Q) In Figure legends, Instead of writing Picture A ; the author can write A only and when the author is referring to it they can refer it as Fig XA where X is a number ( eg 4A, 6A, 9A, etc).

We have changed it accordingly.

  1. Q) In result section lines 251-253 “A total of 2228 micronodules were detected, the majority of which (1924) were found in rhesus macaques, while 304 micronodules were detected in cynomolgus (Table 2).” But table 2 suggest 305 micronodule in cynomolgus (CUV+CIDV ) and (RUV+RAEV+RIDV) 1927 in rhesus . Can the author clarify?

Yes, it is 304 micronodules for cynomolgus and 1927 in Rhesus. There is a concordance between the table 2 and the text.

  1. Q) Line 393 Figure 10 is missing.

We have added it. Sorry.

  1. Q) In the Discussion author has given a general statement that “ Whilst this mucosal aerosol vaccination was better able to reduce the number of isolated micronodules, potentially because of a better capacity to reach and stimulate the immune response in the whole lung “ Although, for rhesus macaques author has taken both conditions (aerosol BCG-vaccinated; intradermal BCG-vaccinated) but not for cynomolgus. 

The work reported here describes new and novel analysis of CT scans collected from macaques enrolled in a number of studies performed to address different questions. The number of macaques that can be used in a single study is limited because of many factors that include ethical considerations, availability, specialist facilities, cost etc. All studies included experimental infection with the strain of Mycobacterium tuberculosis following aerosol exposure to an equivalent dose thus the combination of data across studies provided a larger data set for interrogation. Similarly, BCG vaccination was conducted with the same strain, dose, when used, we have not evaluated the efficacy of aerosol BCG in cynomolgus macaques so do not have CT scans that would allow comparison with the data from rhesus macaques

  1. Q) Why the author has not included the group “ cynomolgus with aerosol BCG vaccinated”?

There were a lot of circumstances: mainly lack of funding.

  1. Q) I table 3 Under the EG category, What author mean by n?

n means number of lesions. We have added the explanation in the legend.

  1. Q) What advantages and drawbacks of using these measures as a new tools to evaluate disease progression? Include it in the discussion

We have included a sentence at the end of the summary: “Equally, the use of data across studies increased the power of the analysis applied and obtained more information without requiring the use of further macaques in line with the principles of the 3R’s”. 

Reviewer 2 Report

The manuscript is trying to correlate daughter micronodules development and Mtb infection by analyzing CT imaging collected from cynomolgus and rhesus macaques infected with Mtb and treated with or without vaccine. The resulted are well organized. But I am wondering whether the table 1 and 2 are showed necessarily and properly. Because the authors summarize all the data points while the infection progress differently and the number of each group is also variable.

Author Response

The manuscript is trying to correlate daughter micronodules development and Mtb infection by analyzing CT imaging collected from cynomolgus and rhesus macaques infected with Mtb and treated with or without vaccine. The resulted are well organized. But I am wondering whether the table 1 and 2 are showed necessarily and properly. Because the authors summarize all the data points while the infection progress differently and the number of each group is also variable.

Thank you very much for the comment. We think both tables are necessary as a summary on the presence of the different lesions. The different progression per time point has been also displayed in other figures, but at the end we have found necessary to summarize in order to have enough statistical power to compare them (table 2); and to show a quick view on the findings, which can help the reader to better understand the work done (table 1).

Reviewer 3 Report

Minor comments

In this research article, the authors analyzed CT images collected from cynomolgus and rhesus macaques following exposure to ultrlow dose Mycobacterium tuberculosis (Mtb) aerosols and monitored them for 16 weeks to evaluate the impact of prior intradermal or inhaled BCG-vaccination on the progression of lung disease. The manuscript is well written, and the story obtained from these studies is potentially interesting, I suggest minor changes such as.

Point 1: The experimental method part is clear; however, it would be better to draw a detailed experimental workflow, including (immunization and infection schedule) added as figure 1.

Point 2: It would be nice to add details how many CFUs of Mtb used for NHP models infection or BCG immunizations.

Point 3: There are a few minor grammar mistakes that need to revise it thoroughly.

Point 4: Would be better to add some methodological information in case NHP infection models are divided into two groups Latent and active TB.

Point 5: It would be interesting to show whether there is an impact in the long-term, including in the establishment micronodules, future long-term Mtb infection and protective response experiments would be informative. Overall, I could not really fault the experiments or the interpretation.

Good Luck!

Author Response

Point 1: The experimental method part is clear; however, it would be better to draw a detailed experimental workflow, including (immunization and infection schedule) added as figure 1.

We have included the Figure 1 as requested. Besides we have added more information at M&M “animals”, which complements the supplementary Figure 1, on the time points analyzed per each animal, which reflects our work.

Point 2: It would be nice to add details how many CFUs of Mtb used for NHP models infection or BCG immunizations.

We have added accordingly in the M&Methods “Animals”.

Point 3: There are a few minor grammar mistakes that need to revise it thoroughly.

We have reviewed the grammar mistakes.

Point 4: Would be better to add some methodological information in case NHP infection models are divided into two groups Latent and active TB.

The studies reported describe individuals with active TB that was either controlled during the study period or progressed to levels that met humane endpoint criteria. TB was not latent in any of the subjects.

Point 5: It would be interesting to show whether there is an impact in the long-term, including in the establishment micronodules, future long-term Mtb infection and protective response experiments would be informative. Overall, I could not really fault the experiments or the interpretation.

It is true, but we have not this information so far.

Reviewer 4 Report

In this study, Isabel Nogueira et al. evaluate the progression of Mtb infection in experimental NHP models through the spatial monitoring of the micronodules and the identification of consolidations with daughter micronodules as a tool to evaluate the efficacy of new vaccines. They developed a new method to evaluate outcome in experimental models of TB in NHP based on CT images, which would fit a future machine learning approach to evaluate new vaccines. This study is very interesting and provides a new tool for evaluating vaccine efficacy. However, I have some suggestions and comments to this submission.

Major issues:

1. Animal ethics need to be added in the materials and methods section. Is the study approved and supervised by the ethics committee, and what is the approval number?

2. The writing of this article must be improved. The section of Materials and Methods should describe the methods and materials used in this study rather than the results or a further discussion. For example, “Further CT scans inspection showed the lung lobes to be separated by pleural fissures as well as bronchovascular bundles, thus presenting the same airway anatomy pattern seen in humans”, this section should be moved to the section of Results.

3. The authors claimed that CT scans were evaluated by a medical consultant radiologist with expertise in respiratory diseases blinded to vaccination status and clinical data. I think this work should be completed by at least two radiologists, and the discordant results should be determined by consultation or by consulting another expert. Otherwise, there will be human mistakes or even errors in CT imaging analysis, leading to impaired reliability of the results.

4. There are two main lines of research in this study, one is to evaluate the differences in pulmonary granulomas and other lesions after BCG vaccination in monkeys with different vaccination routes, and the other is the value of CT imaging in evaluating the efficacy of BCG. The author has no focus, and the two story lines are confusing, which can't let the reader grasp the story.

5. In Figure 7, although the fraction of daughter micronodules of five groups were showed, but we do not know the differences of number of daughter micronodules among five groups. In other words, the number of daughter micronodules collected from CT images did not show the protective efficacy of BCG vaccination. There are similar problems in Figure 9.

Minor issues:

1. The length of Introduction section is too long; I suggest that authors should short it and highlight key points.

2. The English language should be improved by an English native-speaker. For example, Mycobacterium tuberculosis has been abbreviated as Mtb in line 44, but Mycobacterium tuberculosis still was used in line 53. The full name can be used for the first time, and the abbreviation is usually used for the second time.

3. The text and numbers in figures 1 to 3 are too small to be legible. It is suggested that the author increase the font size to improve the image resolution.

4. Special symbol in Figure S1 should be explained in the figure legend. Does it indicate a death of a macaque?

5. In Figure S2, the volume of lung collected from cynomolgus and rhesus were compared. Why did the authors compare this data? There are already racial differences in body size and organ size between the two animals, and I do not think it is meaningful to compare lung volume or size between the two animals.

6. Table 1 did not give detailed information. What does the “n=320” mean? What does the “Median” indicate?

Author Response

Major issues:

1.-Animal ethics need to be added in the materials and methods section. Is the study approved and supervised by the ethics committee, and what is the approval number?

The study designs and procedures were approved by the organisational Animal Welfare and Ethical Review Body, [at the time the work was conducted this was Public Health England and now UK Health Security Agency] after which the work is authorised  at a national government level by the UK Home Office under a Project licence. This information has been added to the ‘Animal’ section in the matherials and methods together with further information describing housing and welfare.

  1. The writing of this article must be improved. The section of Materials and Methods should describe the methods and materials used in this study rather than the results or a further discussion. For example, “Further CT scans inspection showed the lung lobes to be separated by pleural fissures as well as bronchovascular bundles, thus presenting the same airway anatomy pattern seen in humans”, this section should be moved to the section of Results.

We have reviewed this section and split the text requested together with the data included in the description of Micronodules, consolidation, cavitation and pleural distance that being a result of the study, have been reinstalled in the Results chapter.

  1. The authors claimed that CT scans were evaluated by a medical consultant radiologist with expertise in respiratory diseases blinded to vaccination status and clinical data. I think this work should be completed by at least two radiologists, and the discordant results should be determined by consultation or by consulting another expert. Otherwise, there will be human mistakes or even errors in CT imaging analysis, leading to impaired reliability of the results.

That is true. “CT scans were evaluated by a medical consultant radiologist with expertise in respiratory diseases and her supervisor. Both were blinded to vaccination status and clinical data. Data shows the consensus between both”.

  1. There are two main lines of research in this study, one is to evaluate the differences in pulmonary granulomas and other lesions after BCG vaccination in monkeys with different vaccination routes, and the other is the value of CT imaging in evaluating the efficacy of BCG. The author has no focus, and the two story lines are confusing, which can't let the reader grasp the story.

We have clarified the story at the end of the introduction: “The aim of this study has been to evaluate the progression of Mtb infection in experimental NHP models through the spatial monitoring of the micronodules. We have found that the identification of consolidations associated with daughter micronodules would be a useful tool to evaluate the efficacy of new vaccines”.

  1. In Figure 7, although the fraction of daughter micronodules of five groups were showed, but we do not know the differences of number of daughter micronodules among five groups. In other words, the number of daughter micronodules collected from CT images did not show the protective efficacy of BCG vaccination. There are similar problems in Figure 9.

The number of daughter micronodules per each experimental group is displayed in the Table 2. Figure 9 shows the anatomical Distribution of the lesions in vaccinated macaques. This Picture is not intended to discern between the type of micronodules, as it has a low resolution.

Minor issues:

1.-The length of Introduction section is too long; I suggest that authors should short it and highlight key points.

We have shortened the introduction and highlight the key points.

  1. The English language should be improved by an English native-speaker. For example, Mycobacterium tuberculosis has been abbreviated as Mtb in line 44, but Mycobacterium tuberculosis still was used in line 53. The full name can be used for the first time, and the abbreviation is usually used for the second time.

The paper has been reviewed again by an English native-speaker, and the problem with the abbreviations fixed.

  1. The text and numbers in figures 1 to 3 are too small to be legible. It is suggested that the author increase the font size to improve the image resolution.

We have increased the font size in the numbers of these figures.

  1. Special symbol in Figure S1 should be explained in the figure legend. Does it indicate a death of a macaque?

We have included an explanation in the legend: “Macaques marked with dagger symbol were euthanized before the planned end of study, according to the welfare monitoring plan”.

  1. In Figure S2, the volume of lung collected from cynomolgus and rhesus were compared. Why did the authors compare this data? There are already racial differences in body size and organ size between the two animals, and I do not think it is meaningful to compare lung volume or size between the two animals.

Even when this is obvious, as we had to build “in silico” lungs per each macaque, we just wanted to show that effectively our data reflects what is already known. It is just a validation and we think it is worthy to add it in the supplementary file.

  1. Table 1 did not give detailed information. What does the “n=320” mean? What does the “Median” indicate?

We have included an explanation in the legend of Table 1: “n means the total number of lesions per each experimental group. Median shows the value separating the higher half from the lower half of the number of each type of lesion per experimental group”.

Round 2

Reviewer 4 Report

I have checked the revised manuscript carefully, and found that the authors have addressed all my conserns.
I would like to recommend to accept it in this version.